# Impact of altitude on COVID-19 infection and death in the United States: A modeling and observational study

Kenton E. Stephens[1], Pavel Chernyavskiy[2], Danielle R. Bruns[1,3]*

1 WWAMI Medical Education, University of Washington School of Medicine, Seattle, Washington, United States of America, 2 Department of Mathematics and Statistics, University of Wyoming, Laramie, Wyoming, United States of America, 3 Division of Kinesiology & Health, University of Wyoming, Laramie, Wyoming, United States of America

* dbruns1@uwyo.edu

## Abstract

### Background

COVID-19, the disease caused by SARS-CoV-2, has caused a pandemic, sparing few regions. However, limited reports suggest differing infection and death rates across geographic areas including populations that reside at higher elevations (HE). We aimed to determine if COVID-19 infection, death, and case mortality rates differed in higher versus low elevation (LE) U.S. counties.

### Methods

Using publicly available geographic and COVID-19 data, we calculated per capita infection and death rates and case mortality in population density matched HE and LE U.S. counties. We also performed population-scale regression analysis to investigate the association between county elevation and COVID-19 infection rates.

### Findings

Population density matching of LA (< 914m, n = 58) and HE (>2133m, n = 58) counties yielded significantly lower COVID-19 cases at HE versus LE (615 versus 905, p = 0.034). HE per capita deaths were significantly lower than LE (9.4 versus 19.5, p = 0.017). However, case mortality did not differ between HE and LE (1.78% versus 1.46%, p = 0.27). Regression analysis, adjusted for relevant covariates, demonstrated decreased COVID-19 infection rates by 12.82%, 12.01%, and 11.72% per 495m of county centroid elevation, for cases recorded over the previous 30, 90, and 120 days, respectively.

### Conclusions

This population-adjusted, controlled analysis suggests that higher elevation attenuates infection and death. Ongoing work from our group aims to identify the environmental, biological, and social factors of residence at HE that impact infection, transmission, and

**Data Availability Statement:** Data are available in GitHub: https://github.com/pchernya/Covid_Elev.

**Funding:** This work was supported by Wyoming-WWAMI and NIH/NIA K01 AG058810 (DRB). The funders had no role in study design, data collection

and analysis, decision to publish, or preparation of the manuscript.

**Competing interests:** The authors declare no competing interests.

pathogenesis of COVID-19 in an effort to harness these mechanisms for future public health and/or treatment interventions.

## Introduction

Coronavirus disease 2019 (COVID-19) is an illness caused by novel coronavirus Severe Acute Respiratory Syndrome Coronavirus 2 (SARS-CoV-2) that emerged in Wuhan, China in late 2019 and has rapidly proceeded to cause a pandemic. COVID-19 has reached nearly every corner of the globe and affects every demographic. While individual risk factors such as age, male sex, hypertension, diabetes, and heart disease have been previously identified [1], COVID-19 also appears to have unequal infection rates and mortality across geographical regions, suggesting that a combination of social, environmental, and biological risk factors may affect transmission, infection, morbidity, and mortality. One such environmental factor which has attracted interest over the past few months and which our group has significant interest and expertise, is high altitude residence.

The first report of high altitude regions and COVID-19, published in April 2020, demonstrated lower infection rates and mortality in high altitude regions in Tibet, Peru, and Ecuador in comparison to low-altitude regions in the same countries [2]. Since then, subsequent reports have shown no impact of altitude on infection [3], attenuated infection at high altitude [4,5], while others have suggested that COVID-19 infection rates are lower at high altitude, but mortality is not [6], and worsened mortality in high altitude regions [7]. These conflicting findings are likely due to differences in population density, lack of control for population density in the statistical model, as well as due to limited reports of cases and deaths in sparsely populated and/or remote high altitude locations. Approaching the hypothesis that high altitude residence is protective against COVID-19 may require multiple quantitative epidemiology perspectives. We set out to test the hypothesis that higher elevations (HE) attenuates COVID-19 infection rates using two distinct approaches: matching of HE and low elevation (LE) regions and using a statistical modeling approach. We performed U.S. county-level regression analysis, which allowed us to examine the contribution of county centroid elevation in the presence of other risk factors and county-specific latent spatial effects. In an effort to reflect the temporal dynamics of the pandemic, we evaluated the association of county centroid elevation with incidence recorded over the previous 30, 90, and 120 days. To our knowledge, this is the first systematic, population-density adjusted epidemiologic investigation of the impact of altitude on COVID-19 disease infection, deaths, and mortality rates in the U.S. Further, we discuss the biological, social, and environmental contributors to COVID-19 infection, transmission, and pathogenesis, and how these factors are impacted by residence at high altitude.

## Methods

### Matching of high and low altitude counties

High elevation was defined as average elevation greater than 2,133m above sea level. Low elevation was defined as less than 914m. This distinction purposefully omitted locations of moderate elevation, facilitating robust comparisons. Previous reports of COVID and altitude have utilized definitions of higher than 2,800m; however, in the U.S., only 14 counties have an average elevation above this cutoff. Therefore, to reduce statistical noise in the data and due to known clinical and physiological impact of altitude on human physiology at and above 2,000m [8], we used a less restrictive definition. A total of 58 counties above 2,133m were identified. We then matched

58 LE counties as controls. HE and LE counties were matched based on population densities: each HE county was matched to a LE county with a population density between 0.75 and 1.25 times the density of the HE county (0.75 < LE population density/HE population density < 1.25). Population of each county was obtained from federal estimates for the 2019 population based on the 2010 U.S. Census. County area was gathered from the U.S. Census Bureau's 2011 compendium of county areas in square miles. Population density was defined as 2011 population divided by county area in square miles, yielding a density measured in persons per square mile.

COVID-19 cases and deaths were obtained from state and county public health department websites at the end of the business day (AKDT time) on 8/7/2020. Per capita infection rates were calculated using the formula county COVID-19 cases divided by the estimated 2019 county population and then multiplying by 100,000. Per capita death rates were calculated in an identical manner. County COVID-19 mortality rates were calculated by dividing the cumulative county deaths due to COVID-19 by cumulative COVID-19 infections. In cases where counties reported zero COVID-19 cases, the case mortality was set equal to zero.

Infection, death, and case mortality rates were analyzed by one-sided Student's t-test, with an alpha level of 0.05. Significance of correlation coefficients were also calculated. Data are presented as means ± standard error of the mean.

### Statistical modeling

In addition to the matched case-count analysis, we estimated U.S. county-level regression models, with COVID-19 incidence rates as the outcome. To account for different temporal patterns during the pandemic, we performed our analysis on cases collected over 30, 90, and 120 days prior to 8/27/2020. Case counts were collected from the COVID-19 Data Repository by the Center for Systems Science and Engineering (CSSE) at Johns Hopkins University [9], which are updated daily. Our data covers counties in the 48 contiguous U.S. states and the District of Columbia. County population at-risk and the number of households were obtained from the decennial U.S. census.

As a proxy for residence at high altitude, we computed the elevation in meters of county centroids (geometric centers) using the elevatr software package [10] in R 4.0 [11]. Although centroids do not necessarily represent the location of population within each county, a map of county centroids accurately reflects the variability in elevation around the United States (S1 Fig). Centroid elevations range from -71m to 3401m, with mean elevation of 430m and median elevation of 274m. In addition to county centroid elevation and total county population as an offset term, each model contained the following covariates: 1) 9-level USDA 2013 Rural-Urban Continuum Codes (USDA RUCC), where higher-numbered categories indicate increasingly rural environments; 2) average number of persons per household, computed as county population/number of households; 3) interaction of persons per household and USDA Rural-Urban Continuum Codes; 4) independent state random effects; and 5) spatially-correlated county random effects. All estimation was performed using Negative-Binomial and Tweedie-Poisson Generalised Additive Models within the mgcv R package [12]. For more details about the statistical analysis, the reader is directed to Supplementary Statistical Methods. Data sets for analysis and additional S1 File, including R code to reproduce our analyses, are available at https://github.com/pchernya/Covid_Elev.

## Results

### High and low altitude county matching results

Of 3,141 counties in the United Sates, 58 counties from eight states were identified as HE. 58 LE counties of matched population density from 20 states were chosen for comparison. HE

**Table 1. Matched high and low altitude county demographics.**

| Low Elevation (n = 58) | Elevation (m) | Population | Density (persons/mile$^2$) |
|---|---|---|---|
| Maximum | 903 | 485,493 | 775 |
| Minimum | 0 | 654 | 0.51 |
| Mean | 366 | 31,398 | 37.6 |
| High Elevation (n = 58) | | | |
| Maximum | 3,425 | 582,881 | 762 |
| Minimum | 2,141 | 728 | 0.5 |
| Mean | 2,536 | 38,603 | 37.3 |

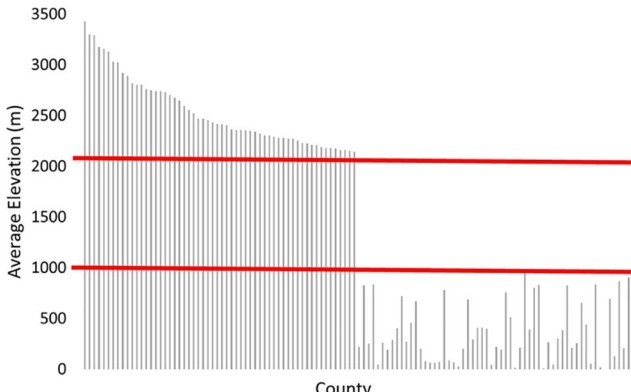

**Fig 1. High and low altitude counties by elevation (m).** Red lines mark the definition of high altitude (>2133m) and low altitude (<914m).

and LE county matching by population density and discrimination by altitude was successful (Table 1). Graphical representation of county altitudes is shown in Fig 1.

Average HE county COVID-19 infection rate was 615 ± 71 cases per 100,000 population, which was statistically significantly lower than the LE county average infection rate of 907 ± 141 (p = 0.034; Fig 2A). Infection rate did not correlate with population density (r = 0.01 and 0.22 for either HE or LE, Fig 2B). Average HE county COVID-19 death rate were 9 ± 2 per 100,000 population and statistically significantly lower than average LE county 19 ± 4 (p = 0.017; Fig 2C). Deaths per 100,000 showed a positive association with population density at both HE and LE locations (r = 0.26 and 0.26, respectively). Even when removing HE (n = 27) and LE (n = 31) counties with infection and death counts of zero, per capita infection and death rates remained significantly lower at HE (p = 0.002 and p = 0.001; S2 Fig). Average COVID-19 case mortality between HE and LE counties was not statistically significantly different at 1.78 ± 0.2% and 1.46 ± 0.4%, respectively (p = 0.267; Fig 2C).

## Modeling results

Our models explain a large portion of variability in COVID-19 incidence with model-based estimates of 80.2%, 84.0%, and 79.7%, for models fit to 120-day, 90-day, and 30-day case counts, respectively. Adjusted for other covariates, incidence rates decreased by 11.72% (16.07%, 7.14%), 12.01% (16.10%, 7.72%), and 12.82% (17.18%, 8.23%) per 495 meters of elevation on average, for cases recorded over the previous 120, 90, and 30 days, respectively. To investigate the potential of a non-linear association between elevation and incidence, we fitted models that included a general smooth function of elevation for each of the three outcome

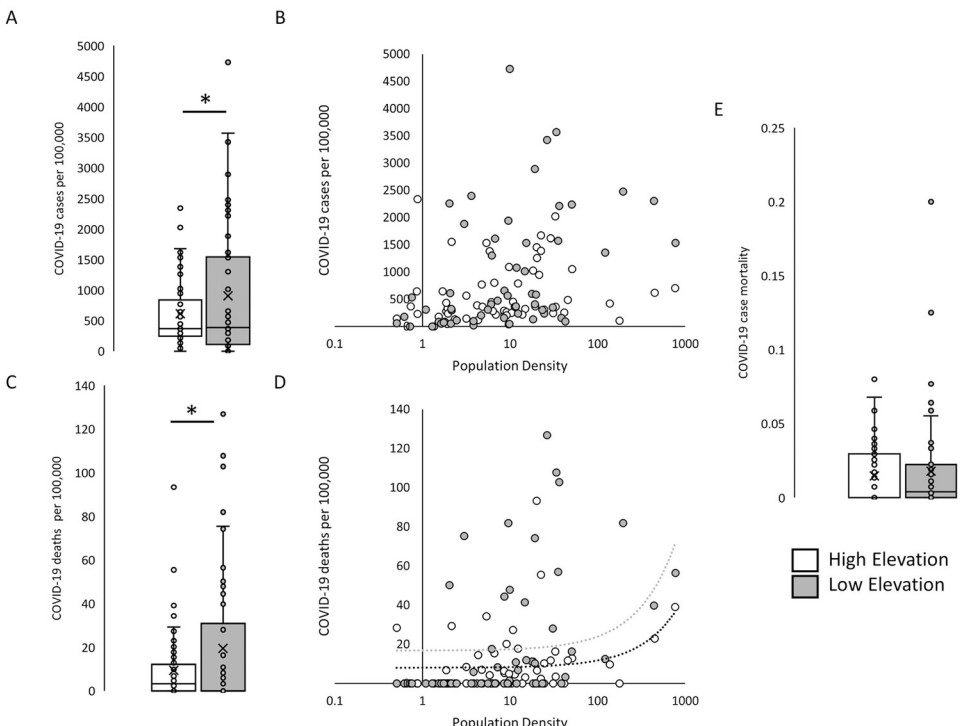

**Fig 2. COVID-19 infection and death in matched high and low elevation counties.** A) Mean COVID-19 cumulative per capita incidence per 100,000 population B) and lack of correlation with population density. C) Mean COVID-19 cumulative per capita death per 100,000 population D) positively correlated with population density at both high- and low-elevation counties. E) COVID-19 case mortality in high and low elevation counties of similar population density. N = 58 for both high elevation and low elevation counties. *p<0.05 by one-sided t-test.

variables. For incidence over 120 days (Fig 3A) and 90 days (Fig 3B), the relationship with elevation is approximately linear from the minimum elevation through the point 3SD above mean elevation (1915m). For counties with centroids above 1915m, uncertainty grows and the strength of association between elevation and incidence diminishes. For incidence over 30 days (Fig 3C), evidence of a non-linear relationship is weak and a model with a linear relationship between elevation and incidence is preferred (S1 Table).

## Discussion

The COVID-19 pandemic has rapidly reached countries and individuals from all demographics. However, some preliminary reports suggest that environmental factors such as altitude may impact disease infection and pathogenesis. To test the hypothesis that residence at high altitude attenuates disease infection and outcome, we used publicly available data to determine infection, death, and case mortality rates in U.S. counties of high and low altitude. For the first time, we provide rigorously population-matched and altitude-delineated analysis of COVID-19 outcomes. Cumulative per capita COVID-19 incidence and death rates were significantly lower in HE counties in comparison to LE counties, with similar case mortality rates. Additionally, we offer complimentary evidence in favor of our hypothesis using a county-level regression model. To our knowledge, our model is the first to examine the effects of elevation in the presence of state and county effects that control for latent legislative, environmental, and social risk factors, such as adherence to public health guidelines, population demographics, and attributes of the built environment. Our analysis suggests that elevation is inversely

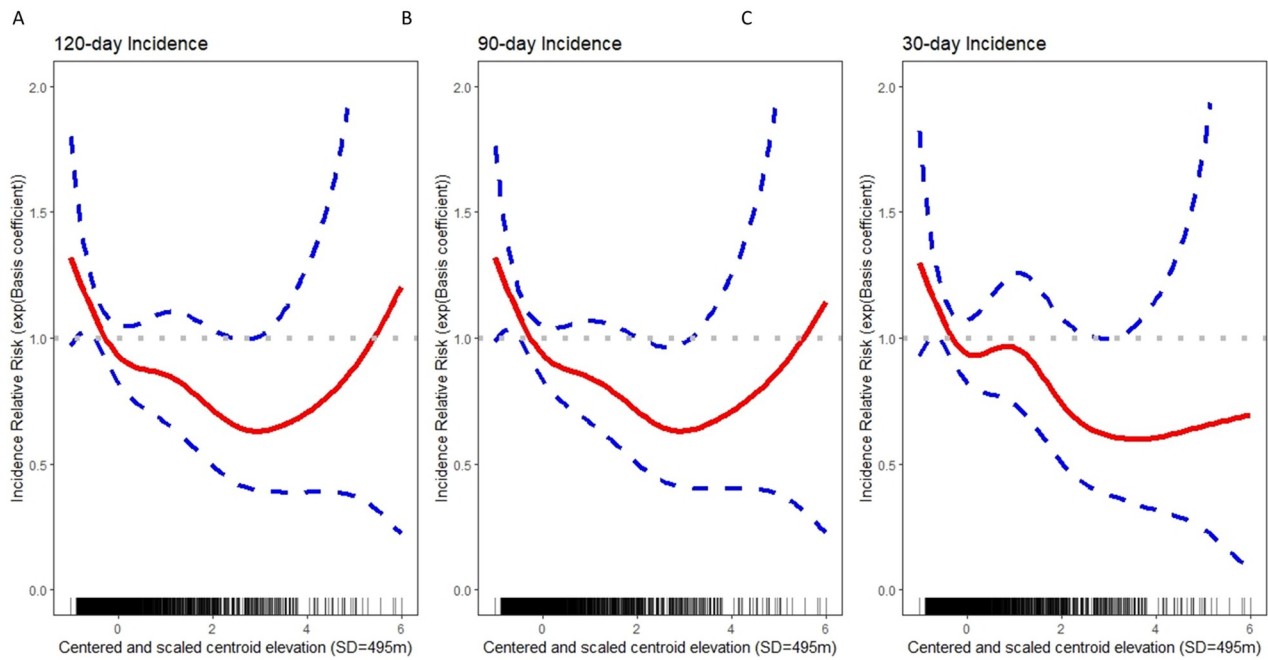

**Fig 3. U.S. county-level regression models, with incident COVID-19 cases as the outcome.** A) 120-day incidence decreased by 11.72% (16.07%, 7.14%) on average, B) 90-day incidence by 12.01% (16.10%, 7.72%) on average, and C) 30-day incidence decreased by 12.82% (17.18%, 8.23) on average per 495 meters of elevation on average, after adjustment for covariates.

associated with incidence through an elevation of 1915m (i.e., 3SD above mean elevation), with little-to-no association as elevation increases further. Two of our models suggest a slight increase in risk at elevations higher than 1915m, which may be an artifact of sparsity in the data, or the fact that many tourist destinations (e.g., ski resorts) that experienced early COVID-19 outbreaks tend to be located in those counties.

Together, these analyses offer evidence that residence at HE attenuates SARS CoV-2 infection and thus death rates, without altering disease pathogenesis such that once infected, risk of death (mortality) is similar at HE versus LE. The differences in COVID-19 outcomes at higher elevations are likely multifactorial and affect transmission, infection, and pathogenesis (Fig 4). Furthermore, these factors likely stem from environmental, biological, as well as social and policy-level differences. Future efforts aimed at understanding these factors and how they differ in populations of residence at higher elevations are critical for modification of disease outcomes.

## SARS-CoV-2 transmission at high altitude

Transmission of SARS-CoV-2 occurs through respiratory droplets, aerosols, and fomites [13]. Several factors impact viral survival outside of the host, including temperature, humidity, and type and intensity of UVB light. SARS-CoV-2 infectivity is attenuated with increasing temperature and humidity [14] as well as with higher intensity of UVB [15]. At higher altitudes, temperature and relative humidity decrease, while intensity of UVB light increases [16]. On the other hand, greater sunlight leads to higher vitamin D, which increases host T-cell mediated protection to viral pathogens [17]. Limited and discrepant reports suggest that temperature, humidity, vitamin D, and UV light contribute modestly to SARS-CoV-2 transmission [18–21], but these reports lack consensus and to our knowledge have not been

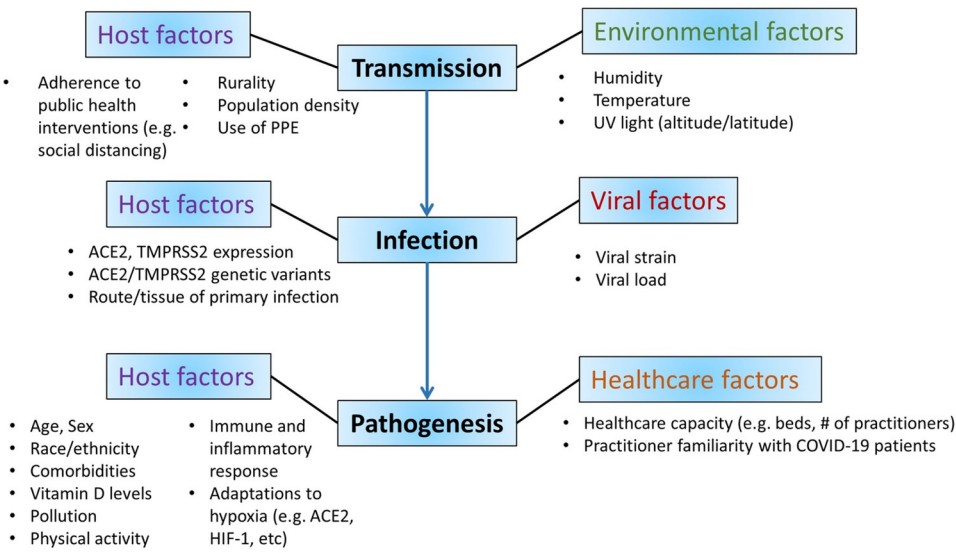

**Fig 4. Summary of proposed mechanisms of difference in COVID-19 infection, transmission, and pathogenesis at high altitude.** Factors which may impact COVID-19 infection and death at high altitude include host, environmental, viral, and healthcare factors.

extrapolated to specific geographic locations such as altitude. Since many environmental variables change with altitude and their respective effects on SARS-CoV-2 infectivity can be at odds, the sum effect of increasing altitude on COVID-19 transmission is complex and requires further study to determine which environmental factors have the greatest effect on viral transmission.

## SARS-CoV-2 infection at high altitude

The interaction between infectious agent and host has long been known to be impacted by environmental factors. Hypoxia was among the first such studied environmental factors, with reports in the early 1950's that mice housed with reduced atmospheric pressure at simulated altitude demonstrated attenuated viral load and mortality [22,23]. With specific regard to SARS-CoV-2, expression of ACE2, the receptor used for cellular entry by SARS-CoV-2 [24,25] is altered in response to hypoxia or simulated altitude. However, these data are limited and inconsistent. While some reports found decreased in ACE2 expression in human pulmonary artery smooth muscle cells placed in a hypoxic environment (2% $O_2$) for 12 days [26] and decreased ACE2 in right ventricular tissue from rats in hypobaric hypoxia (4800 m) for 28 days [27], others have reported ACE2 upregulation in hypoxic environments [28,29]. In addition to conflicting tissues, altitude/hypoxia exposure, and outcomes, none of these pre-clinical mechanistic investigations have utilized models which reside at altitude, but rather have instead utilized short-term ascent to altitude. Thus, there is a critical need to understand the impact of chronic hypoxia/high altitude on ACE2 expression as a mechanism of SARS-CoV-2 infection. Preliminary data from our lab suggests that mice that reside at HE (1915m) for several generations have significantly lower cardiac ACE2 expression than animals raised at sea level. Further study of ACE2 expression in the lungs, respiratory tract, and in other tissues is essential to determining if and to what degree ACE2 expression (and other SARS-CoV-2 receptors) changes in response to hypoxia (altitude), and if this confers susceptibility or protection to COVID-19 infection.

## Pathogenesis of SARS-CoV-2 and COVID-19 at high altitude

Our data demonstrate that HA-mediated attenuation of infection results in lower death rates with similar case mortality, implying that following infection, disease pathogenesis or severity is not attenuated by elevation. However, even in locations where COVID-19 testing is adequate, asymptomatic cases are likely largely undetected. Thus, it is possible that residence at higher elevation favors asymptomatic infection, due to host adaptation, presence of comorbidities, or other factors. COVID-19 has disproportionately affected people of color, the elderly, and those with pre-existing conditions such as diabetes and obesity [1]. Residents of HE are less likely to be obese than lowlanders [30], likely both through physiological regulation of metabolism at hypoxia, but also due to lifestyle and social factors such as physical activity. Demographic reports of high versus low altitude locations are warranted, given some suggestion that older individuals, especially those with comorbid conditions, are likely to relocate to lower altitude due to reasons of poor health [31]. On the other hand, host adaptations to living at and side effects of altitude may provide protection against COVID-19 disease. Such defenses include attenuation of comorbidities as discussed above, tolerance to hypoxemia given lower $O_2$ saturations at altitude, and others. Whether these host factors impact infection risk also remain unknown and likely could contribute to the attenuated infections observed in our study. Lastly, environmental factors which impact pathogenesis likely contribute to outcomes, such as air quality. Pollution is a proposed susceptibility factor for COVID-19 [32] and may differ at higher elevations, especially during wildfire season in the western United States. Together, it is clear that etiology, pathogenesis, and outcomes of COVID-19 differ at higher elevation locations, likely a result of complex interplay between these factors.

Our study has several limitations. We report cumulative as well as 30, 90, and 120-day infection rates as of late August 2020. As such, our data largely reflect the early pandemic and the summer months in the United States. Given the known association between seasonality and COVID-19 transmission, we suggest that future work elucidate the interaction between season and high altitude. Elevation of county centroids only serves as a proxy variable and does not necessarily reflect the elevation of where residents reside in any given county. Our data is drawn from population-level data sources and we do not have access to individual-level data. Individual-level analyses would permit control for comorbid conditions, COVID-19 hospitalizations and complications, and would almost certainly yield insight into potential mechanisms by which altitude affects infection and survival. Analyses of the targeted mechanisms by which altitude protects against COVID-19 infection are critical for understanding of SARS-CoV-2 infection, transmission, and pathogenesis, and for design of interventions which attenuate poor outcomes. While we demonstrate that residence at higher elevations is protective against COVID-19 infection and death, we also caution that these data are associations drawn from residents of altitude, rather than acute ascent. We strongly suggest that future work is needed to understand how high altitude impacts COVID-19. Public health guidance and preventive measures must continue to be practiced by high altitude residence and visitors.

## Supporting information

**S1 Fig. Unprojected choropleth map of U.S. county centroids colored by elevation in meters.** Elevation patterns based on county centroids closely represent elevation patterns of the continental U.S.
(DOCX)

**S2 Fig. COVID-19 infection and death in matched high and low altitude counties with removal of counties with infection and death counts of zero.** A) Mean COVID-19

cumulative per capita incidence per 100,000 population. B) Mean COVID-19 cumulative per capita death per 100,000 population. C) COVID-19 case mortality in high and low altitude counties of similar population density. N = 33 for high altitude and N = 26 for low altitude counties. *p<0.05 by one-sided t-test.
(DOCX)

**S1 Table. Akaike Information Criterion (AIC) and percent deviance explained in parentheses for the statistical models considered.** Smaller AIC and larger percent deviance explained constitutes the preferred model.
(DOCX)

**S1 File.**
(DOCX)

## Acknowledgments

The authors thank Tim Robinson and Emily Schmitt.

## Author Contributions

**Conceptualization:** Kenton E. Stephens, Danielle R. Bruns.

**Data curation:** Kenton E. Stephens, Pavel Chernyavskiy, Danielle R. Bruns.

**Formal analysis:** Kenton E. Stephens, Pavel Chernyavskiy, Danielle R. Bruns.

**Funding acquisition:** Danielle R. Bruns.

**Investigation:** Kenton E. Stephens, Pavel Chernyavskiy, Danielle R. Bruns.

**Methodology:** Pavel Chernyavskiy.

**Visualization:** Kenton E. Stephens, Pavel Chernyavskiy, Danielle R. Bruns.

**Writing – original draft:** Kenton E. Stephens, Pavel Chernyavskiy, Danielle R. Bruns.

**Writing – review & editing:** Kenton E. Stephens, Pavel Chernyavskiy, Danielle R. Bruns.

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
