## [Decision Letter · Decision Letter 0]

3 Nov 2020

PONE-D-20-28278

Impact of altitude on COVID-19 infection and death in the United States: a modeling and observational study

PLOS ONE

Dear Dr. Bruns,

Thank you for submitting your manuscript to PLOS ONE. After careful consideration, we feel that it has merit but does not fully meet PLOS ONE’s publication criteria as it currently stands. Therefore, we invite you to submit a revised version of the manuscript that addresses the points raised during the review process.

Please respond to each of the reviewer's comments on a point-by-point basis and revise the manuscript accordingly.

We look forward to receiving your revised manuscript.

Kind regards,

Jeffrey Shaman

Academic Editor

PLOS ONE

Journal Requirements:

Reviewers' comments:

Reviewer's Responses to Questions

**Comments to the Author**

1. Is the manuscript technically sound, and do the data support the conclusions?

Reviewer #1: Yes

Reviewer #2: Partly

Reviewer #3: Yes

2. Has the statistical analysis been performed appropriately and rigorously? 

Reviewer #1: Yes

Reviewer #2: No

Reviewer #3: Yes

3. Have the authors made all data underlying the findings in their manuscript fully available?

Reviewer #1: Yes

Reviewer #2: Yes

Reviewer #3: No

4. Is the manuscript presented in an intelligible fashion and written in standard English?

Reviewer #1: Yes

Reviewer #2: Yes

Reviewer #3: Yes

5. Review Comments to the Author

Reviewer #1: Interesting article that further supports the role of host factors in viral spread in community. Among many of the previously published articles, there is one that found association of lower altitude and population density with lethality rates, please cite it in the discussion: 10.1089/ham.2020.0168 which only included cities above a certain population cut-off.

Another important point omitted in the discussion is that the dataset included the Summer months. As we currently know the host factors in the summer make pathogenicity extremely low compared to winter and early spring. It is ok to include summer months, however, an attenuator is present that is independent of elevation, in the last months of data collection.

Reviewer #2: The incidence of COVID-19 at high altitude is garnering an attention. The rigorous science is required for the conclusion and it has to be with cautionary as the stakes are too high especially when the conclusion tends to be lower incidence. Specific comments:

Authors have take a liberal approach in classification of altitude (in their own words). They must be taking it very seriously and it is the primary objective to map out COVID-19. So they cannot take liberal approach.

I strongly suggest authors to stratify their data in different altitudes: <500m, <1000m, <1500m, <2000m, <2500m, <3000m and >3000m for the incidence analysis.

Secondly, the altitude cut off should be re-arranged <500 as low altitude and high altitude as >2500m. Authors have found a reference that suits them. High altitude is defined >2500m.

Third, authors have to have analysis based on the altitude of residence not based on the county/province. They have to used absolute altitude of residence (born and raised altitude) of the participants (population) for the consideration. Otherwise, this is going to be a noise rather than real scientific finding.

Fourth, authors need to do better job analysing data with several variables or covariates such as population density, age/sex distribution, testing frequency/facilities, sever COVID-19 facilities, community transmission and others.

Fifth, authors have missed several key and seminal publications in this field. I would highly encourage to keep uptodate and discuss their results accordingly. They should also learn from other literature how they have analysed their data and advance in their analyses. Some of the key literature they must not miss are:

Woolcott & Bergman:

https://www.liebertpub.com/doi/full/10.1089/HAM.2020.0098

Castagnetto et al:

https://www.liebertpub.com/doi/full/10.1089/ham.2020.0173

Lin et al:

https://www.liebertpub.com/doi/full/10.1089/ham.2020.0168

Pun et al:

Lower Incidence of COVID-19 at High Altitude: Facts and Confounders

https://www.liebertpub.com/doi/10.1089/ham.2020.0114

Intimayta-Escalante et al:

https://www.liebertpub.com/doi/full/10.1089/ham.2020.0133

Calvo MS:

https://academic.oup.com/ajcn/article-abstract/112/4/915/5901951

Reviewer #3: Associations of COVID-19 cases and deaths with altitude has been the subject of a number of peer-reviewed and pre-print studies since the emergence of the pandemic. While some studies conclude that likelihood of cases and/or deaths from COVID-19 inversely correlates with high altitude, other studies have not reached the same conclusion. Few studies consider population density in their analyses, which may be a relevant parameter to normalize disease susceptibility vs clustering effects.

In line with other studies, the authors do consider population density in high-altitude (HA) vs low-altitude (LA) sites in the US, and find that HA is associated with lower COVID-19 case and death rates, independent of density. This is an interesting finding by itself.

However, in addition to physiological adaptation to HA, many other variables may impact on this apparent association, in particular co-morbidies and ethnicity. While the latter are discussed in the manuscript, it would be important that these parameters are actually analyzed as part of the study, so as to reach conclusions as the role played by these factors in the observed associations.

It would also be desirable that the authors supply the datasets used for their analyses, in a format amenable to reproduction and re-analysis by readers, including reviewers.

6. PLOS authors have the option to publish the peer review history of their article (what does this mean?). If published, this will include your full peer review and any attached files.

Reviewer #1: **Yes: **carlos gustavo wambier

Reviewer #2: No

Reviewer #3: No

---

## [Author Response · Author response to Decision Letter 0]

9 Nov 2020

Dear Editor and Reviewers-

While responding to revisions as requested below and in the process of making our analysis publicly available, we noticed that at the time of our initial model (August 2020), a mistake of omission had been made with respect to New York City area reporting. At the time we pulled this dataset, NYC was not reporting any confirmed cases- clearly a mistake as there must have been some cases in a population of over 8 million residents. In an effort not to misrepresent the dataset, we re-ran all statistics from 8/27 using the now amended dataset, matching the dates stated in the paper. Nothing else changed- we simply reran the model to be more representative. When we did so, our linear associations with COVID incidence per 495m at 120, 90, and 30 days changed by 0.32, 0.33, and 0.56%, respectively. While these confidence intervals changed minimally, they did change in the direction of our hypothesis- of protection at high altitude. We chose to draw the Reviewer and Editors attention to this matter in an effort to be transparent and thorough in our analysis. 

We thank the reviewers for their thoughtful and constructive feedback. We are pleased the reviewers were generally enthusiastic about our work and its impact. We have made efforts to improve our manuscript, as suggested below. We hope that our work is now suitable for publication in PLOS One.

Reviewer #1: Interesting article that further supports the role of host factors in viral spread in community. Among many of the previously published articles, there is one that found association of lower altitude and population density with lethality rates, please cite it in the discussion: 10.1089/ham.2020.0168 which only included cities above a certain population cut-off.

We thank the reviewer for drawing our attention to this publication. We have referenced the work these authors did in discussing population density and altitude in the introduction. 

Another important point omitted in the discussion is that the dataset included the Summer months. As we currently know the host factors in the summer make pathogenicity extremely low compared to winter and early spring. It is ok to include summer months, however, an attenuator is present that is independent of elevation, in the last months of data collection.

We agree that there may be some seasonality in the protective effect of elevation and have added text in the discussion to this effect and agree that further investigation around the presence of a seasonal effect is warranted. 

While a complete update of our data and analysis is outside the scope of the current manuscript, in an effort to investigate how the change in season influenced the apparent protective effect of elevation, we re-estimated our model using cases collected over the previous 90 days as of 11/04/2020. With these updated data, the estimated linear effect (95% Confidence Interval) per 495m (1 SD) of elevation is -9.44% (-13.01%, -5.72%), which is only somewhat attenuated relative to what we originally reported. Qualitatively speaking, these updated results do not change our findings. Therefore, while future work is warranted for the mixed effects of altitude and seasonality, it does not change the significant effect reported here on altitude and COVID-19 infection. 

Reviewer #2: The incidence of COVID-19 at high altitude is garnering an attention. The rigorous science is required for the conclusion and it has to be with cautionary as the stakes are too high especially when the conclusion tends to be lower incidence. Specific comments:

We agree with the review that the stakes are high, especially when our conclusion is that incidence is lower at high altitude. However, our discussion cautions readers to the conclusions drawn here- that they are associative, not causative, and that likely the benefits are only evident with residence at HA, not acute ascent. We have also amended the discussion to encourage implementation and following of public health guidelines.

Authors have take a liberal approach in classification of altitude (in their own words). They must be taking it very seriously and it is the primary objective to map out COVID-19. So they cannot take liberal approach.

Taking a rigorous approach to study the impact of high altitude on COVID-19 does not require a rigorous definition of altitude. In fact, using strict definitions of altitude has limited previous work from drawing conclusions, given the sparse population at high altitude, the lack of control for population density, and the known impact of mild altitude on disease and viral biology. Further, our modeling results support our classification of altitude, since this approach does not require discrete cut-offs, but rather calculated relative infection risk by 500m intervals. Our novel dual approach, consistent finding that HA protects against infection, and limitations of previous work support the approach that a strict cut-off of >2500m is not necessary to understand the impact of HA on disease transmission. A consensus definition of high altitude has yet to be reached- with some groups utilizing 2400m, others 2800, and some 1500m.

I strongly suggest authors to stratify their data in different altitudes: <500m, <1000m, <1500m, <2000m, <2500m, <3000m and >3000m for the incidence analysis.

While this approach has merit, it is not feasible for the paired approach we employed, since sufficient counties with population matching do not exist at these discrete intervals. However, our modeling analysis does take this discrete approach, as the protection of HA on COVID-19 incidence is reported as +/- SD (500m). 

Secondly, the altitude cut off should be re-arranged <500 as low altitude and high altitude as >2500m. Authors have found a reference that suits them. High altitude is defined >2500m.

As discussed above and in the manuscript, we believe these strict cut-offs limit statistical and biological comparisons. We matched HA with LA counties based on population density, given the well-reported associations between population density and viral spread, especially at locations of HA [1]. We were unable to find comparable matches at <500m for all our HA locations, thus we used slightly higher, but still <914m. Further, our modeling analysis uses discrete 500m changes in elevation and demonstrates significant protection (11% for every 500m), even at elevations <2500m.

Third, authors have to have analysis based on the altitude of residence not based on the county/province. They have to used absolute altitude of residence (born and raised altitude) of the participants (population) for the consideration. Otherwise, this is going to be a noise rather than real scientific finding.

The reviewer is correct that absolute residence of birth and residence would be a cleaner statistical approach. However, this data does not exist in the United States. COVID-19 data is not tracked at a municipality smaller than the county-level, nor does any population-level or census data take into account birth and time of residence at HA. We do not have populations of individuals who have resided for generations at HA, such as those in Tibet and Peru. While we acknowledge this limitation in our discussion, our data are still robust enough and our effects still significant enough to capture the effect of HA on COVID-19 infection.

Fourth, authors need to do better job analysing data with several variables or covariates such as population density, age/sex distribution, testing frequency/facilities, sever COVID-19 facilities, community transmission and others.

The reviewer is correct that these covariates are significant as they relate to COVID-19 transmission and mortality. We’ve controlled for rurality, persons per household, interaction of persons per household and rurality, state effects, and spatial effects in our modeling analysis. Unfortunately, some of the other variables such as testing frequency and facilities are not available for analysis. However, we have discussed these variables as contributors in our discussion.

Fifth, authors have missed several key and seminal publications in this field. I would highly encourage to keep uptodate and discuss their results accordingly. They should also learn from other literature how they have analysed their data and advance in their analyses. Some of the key literature they must not miss are:

We thank the reviewer for drawing our attention to these publications, several of which were published after submission of our work. They are all included in our references and discussion.

Woolcott & Bergman:

https://www.liebertpub.com/doi/full/10.1089/HAM.2020.0098

Castagnetto et al:

https://www.liebertpub.com/doi/full/10.1089/ham.2020.0173

Lin et al:

https://www.liebertpub.com/doi/full/10.1089/ham.2020.0168

Pun et al:

Lower Incidence of COVID-19 at High Altitude: Facts and Confounders

https://www.liebertpub.com/doi/10.1089/ham.2020.0114

Intimayta-Escalante et al:

https://www.liebertpub.com/doi/full/10.1089/ham.2020.0133

Calvo MS:

https://academic.oup.com/ajcn/article-abstract/112/4/915/5901951

Reviewer #3: Associations of COVID-19 cases and deaths with altitude has been the subject of a number of peer-reviewed and pre-print studies since the emergence of the pandemic. While some studies conclude that likelihood of cases and/or deaths from COVID-19 inversely correlates with high altitude, other studies have not reached the same conclusion. Few studies consider population density in their analyses, which may be a relevant parameter to normalize disease susceptibility vs clustering effects.

In line with other studies, the authors do consider population density in high-altitude (HA) vs low-altitude (LA) sites in the US, and find that HA is associated with lower COVID-19 case and death rates, independent of density. This is an interesting finding by itself.

However, in addition to physiological adaptation to HA, many other variables may impact on this apparent association, in particular co-morbidies and ethnicity. While the latter are discussed in the manuscript, it would be important that these parameters are actually analyzed as part of the study, so as to reach conclusions as the role played by these factors in the observed associations.

We thank the reviewer for their review of our work and recognition of the work we did. We completely agree that other covariates such as comorbidities and ethnicity are important. We have controlled for several covariates known to impact transmission- rurality, number of persons per household state and spatial effects. Further, we discuss several of these other variables- particularly comorbidities, as a mechanism by which HA and LA differ. Further, we propose that future epidemiological studies utilize individual-level data to begin to understand the complex relationships between factors such as comorbid conditions, ethnicity, and altitude.

It would also be desirable that the authors supply the datasets used for their analyses, in a format amenable to reproduction and re-analysis by readers, including reviewers.

 The datasets used in our analysis are now available at https://github.com/pchernya/Covid_Elev.

 

Supporting References

1. Lin, E.M., A. Goren, and C. Wambier, Letter to the Editor: Environmental Effects on Reported Infections and Death Rates of COVID-19 Across 91 Major Brazilian Cities. High Alt Med Biol, 2020.

---

## [Decision Letter · Decision Letter 1]

30 Nov 2020

PONE-D-20-28278R1

Impact of altitude on COVID-19 infection and death in the United States: a modeling and observational study

PLOS ONE

Dear Dr. Bruns,

Thank you for submitting your manuscript to PLOS ONE. After careful consideration, we feel that it has merit but does not fully meet PLOS ONE’s publication criteria as it currently stands. Therefore, we invite you to submit a revised version of the manuscript that addresses the points raised during the review process.

Note that two of the reviewers were not wholly satisfied with your response to the previous round of comments and highlight particular issues that must be addressed if this manuscript is to be accepted.

We look forward to receiving your revised manuscript.

Kind regards,

Jeffrey Shaman

Academic Editor

PLOS ONE

Reviewers' comments:

Reviewer's Responses to Questions

**Comments to the Author**

1. If the authors have adequately addressed your comments raised in a previous round of review and you feel that this manuscript is now acceptable for publication, you may indicate that here to bypass the “Comments to the Author” section, enter your conflict of interest statement in the “Confidential to Editor” section, and submit your "Accept" recommendation.

Reviewer #1: All comments have been addressed

Reviewer #2: (No Response)

Reviewer #3: (No Response)

2. Is the manuscript technically sound, and do the data support the conclusions?

Reviewer #1: Yes

Reviewer #2: (No Response)

Reviewer #3: Yes

3. Has the statistical analysis been performed appropriately and rigorously? 

Reviewer #1: Yes

Reviewer #2: (No Response)

Reviewer #3: Yes

4. Have the authors made all data underlying the findings in their manuscript fully available?

Reviewer #1: Yes

Reviewer #2: (No Response)

Reviewer #3: Yes

5. Is the manuscript presented in an intelligible fashion and written in standard English?

Reviewer #1: Yes

Reviewer #2: (No Response)

Reviewer #3: Yes

6. Review Comments to the Author

Reviewer #1: The manuscript was improved after the review, and this manuscript represents further evidence of the impact of altitude and all environmental factors changed by altitude in modulating COVID. These information is of great importance for future understanding of host-virus interactions and direct further research.

Reviewer #2: I am not convinced that this paper addresses altitude effect in COVID-19 incidence/mortality when you are compromising definition of altitude or there is no altitude per se. It is like changing goal post or cut-off p-value to show significance. However, I am fine with the content if authors change title and conclusion that effect of altitude. I would suggest something like 'geographical variation' in place of 'altitude.

The literature suggested are not all incorporated/implemented by the authors although they responded they have done that.

Reviewer #3: The authors have not addressed the following request: "In addition to physiological adaptation to HA, many other variables may impact on this apparent association, in particular co-morbidies and ethnicity. While the latter are discussed in the manuscript, it would be important that these parameters are actually analyzed as part of the study, so as to reach conclusions as the role played by these factors in the observed associations."

This could be addressed at least as association analysis of COVID19 incidence/death rates with prevalence of potential co-morbidities.

7. PLOS authors have the option to publish the peer review history of their article (what does this mean?). If published, this will include your full peer review and any attached files.

Reviewer #1: **Yes: **Carlos Gustavo Wambier

Reviewer #2: No

Reviewer #3: No

---

## [Author Response · Author response to Decision Letter 1]

2 Dec 2020

Reviewer #1: The manuscript was improved after the review, and this manuscript represents further evidence of the impact of altitude and all environmental factors changed by altitude in modulating COVID. These information is of great importance for future understanding of host-virus interactions and direct further research.

 We thank the Reviewer for their time and contributions to our work.

Reviewer #2: I am not convinced that this paper addresses altitude effect in COVID-19 incidence/mortality when you are compromising definition of altitude or there is no altitude per se. It is like changing goal post or cut-off p-value to show significance. However, I am fine with the content if authors change title and conclusion that effect of altitude. I would suggest something like 'geographical variation' in place of 'altitude.

Our definition of altitude is not without precedence. We provide citations for why our definition is rigorous. Other publications, including Lin et al, as suggested by the reviewer below, do not use a discrete cut-off of 2,400m/8,000ft. While it’s true that the textbook definition of “high altitude” is a discrete value, a single numeric definition not does not represent biology, nor does it provide a comprehensive picture of how SARS-CoV-2 infection and pathogenicity change with altitude. Our modelling data clearly demonstrate a protective effect of altitude beginning at 1000m above sea level.

We agree that centroid elevation is only a proxy variable; however, we do not have access to residential addresses across the country, for instance, for which we would compute the elevation instead. If this were available, we agree it would serve as a better measure of “residence at high altitude”. However, this level of granularity is largely impractical and would fail to protect the participants’ confidentiality. We plan to continue investigating the effects of residence at high altitude using animal models and hope that this population-based study will serve as substantive motivation for subsequent experiments.

We respectfully disagree with the reviewer’s comment about “geographic variation”. Geographic variation can be captured by models with county and state effects alone, but we would lose the ability to make inferential statements regarding why certain counties are at high risk vs. low risk. By including county and state effects – which capture latent risk - in addition to measurable characteristics, we are able to make inferential statements about these characteristics, like we do with linear and non-linear associations with elevation. 

However, to minimize confusion regarding word choice, we have changed the verbiage of “high altitude” to reference counties of “higher elevation; HE”. We have also amended conclusions to refer to elevation, rather than discrete HA/high altitude. 

The literature suggested are not all incorporated/implemented by the authors although they responded they have done that.

The Reviewer suggested we cite Woolcot&Bergman, Castagnetto Intimayta-Escalante, and Calvo. These references are numbers 7, 3, 4, and 21 in our reference list, respectively. Pun is a review paper, not a primary report. Therefore, while we have not referenced it here, we have referenced many of the references in its citation list. We apologize for the oversight in missing Lin et al. It is now reference 5. 

Reviewer #3: The authors have not addressed the following request: "In addition to physiological adaptation to HA, many other variables may impact on this apparent association, in particular co-morbidities and ethnicity. While the latter are discussed in the manuscript, it would be important that these parameters are actually analyzed as part of the study, so as to reach conclusions as the role played by these factors in the observed associations." This could be addressed at least as association analysis of COVID19 incidence/death rates with prevalence of potential co-morbidities.

We agree that race/ethnicity and the presence of co-morbidities is undoubtedly related to risk of COVID-19, however we feel it would be more epidemiologically meaningful to include those covariates for individual-level data. Because we are working with aggregated data, we initially chose to exclude these covariates to avoid introducing multicollinearity and additional ecological fallacy. 

However, to investigate the Reviewer’s concern, we re-estimated our statistical models for cases recorded over the previous 30, 90, and 120 days as of 11/29/2020, adjusting for percentages of black, hispanic, and asian individuals. These data were collected for each US county during the 2013 5-year American Community Survey. Counties with more minorities tended to have higher incidence rates; however, the inclusion of race/ethnicity only slightly attenuated the linear associations with elevation, which were: -10.6% per 495m for 30-day incidence, -7.7% per 495m for 90-day incidence, and -6.5% per 495m for 120-day incidence. All associations remained statistically significant at an alpha level of 0.01. In addition, models with race/ethnicity explain an additional 1%, 3%, and 4% variability (for 30, 90, 120-day incidence, respectively), relative to what we report in our manuscript (Supplementary Table 1). Taken together, we feel we should keep the statistical models reported in the text targeted towards our stated hypothesis, which covered rurality and density inside the household, but did not cover county demographics.

---

## [Decision Letter · Decision Letter 2]

22 Dec 2020

Impact of altitude on COVID-19 infection and death in the United States: a modeling and observational study

PONE-D-20-28278R2

Dear Dr. Bruns,

We’re pleased to inform you that your manuscript has been judged scientifically suitable for publication and will be formally accepted for publication once it meets all outstanding technical requirements.

Kind regards,

Jeffrey Shaman

Academic Editor

PLOS ONE

Additional Editor Comments (optional):

Reviewers' comments:

Reviewer's Responses to Questions

**Comments to the Author**

1. If the authors have adequately addressed your comments raised in a previous round of review and you feel that this manuscript is now acceptable for publication, you may indicate that here to bypass the “Comments to the Author” section, enter your conflict of interest statement in the “Confidential to Editor” section, and submit your "Accept" recommendation.

Reviewer #2: All comments have been addressed

Reviewer #3: All comments have been addressed

2. Is the manuscript technically sound, and do the data support the conclusions?

Reviewer #2: Yes

Reviewer #3: Yes

3. Has the statistical analysis been performed appropriately and rigorously? 

Reviewer #2: Yes

Reviewer #3: Yes

4. Have the authors made all data underlying the findings in their manuscript fully available?

Reviewer #2: (No Response)

Reviewer #3: Yes

5. Is the manuscript presented in an intelligible fashion and written in standard English?

Reviewer #2: (No Response)

Reviewer #3: Yes

6. Review Comments to the Author

Reviewer #2: I am fine with the changes. The altitude to elevation is not consistent throughout the the manuscript.

Reviewer #3: The authors have provided a reasonable explanation for not conducting in the current study the requested co-morbidity association analyses.

7. PLOS authors have the option to publish the peer review history of their article (what does this mean?). If published, this will include your full peer review and any attached files.

Reviewer #2: No

Reviewer #3: No

---

## [Editor Report · Acceptance letter]

23 Dec 2020

PONE-D-20-28278R2 

Impact of altitude on COVID-19 infection and death in the United States: a modeling and observational study 

Dear Dr. Bruns:

I'm pleased to inform you that your manuscript has been deemed suitable for publication in PLOS ONE. Congratulations! Your manuscript is now with our production department. 

Kind regards, 

on behalf of

Prof. Jeffrey Shaman 

Academic Editor

PLOS ONE